# Underwater CAM photosynthesis elucidated by *Isoetes* genome

David Wickell[1,2], Li-Yaung Kuo [3], Hsiao-Pei Yang[2], Amra Dhabalia Ashok[4], Iker Irisarri[4,5], Armin Dadras [4], Sophie de Vries [4], Jan de Vries [4,5,6], Yao-Moan Huang[7], Zheng Li [8], Michael S. Barker [9], Nolan T. Hartwick [10], Todd P. Michael [10]✉ & Fay-Wei Li [1,2]✉

To conserve water in arid environments, numerous plant lineages have independently evolved Crassulacean Acid Metabolism (CAM). Interestingly, *Isoetes*, an aquatic lycophyte, can also perform CAM as an adaptation to low $CO_2$ availability underwater. However, little is known about the evolution of CAM in aquatic plants and the lack of genomic data has hindered comparison between aquatic and terrestrial CAM. Here, we investigate underwater CAM in *Isoetes taiwanensis* by generating a high-quality genome assembly and RNA-seq time course. Despite broad similarities between CAM in *Isoetes* and terrestrial angiosperms, we identify several key differences. Notably, *Isoetes* may have recruited the lesser-known 'bacterial-type' PEPC, along with the 'plant-type' exclusively used in other CAM and C4 plants for carboxylation of PEP. Furthermore, we find that circadian control of key CAM pathway genes has diverged considerably in *Isoetes* relative to flowering plants. This suggests the existence of more evolutionary paths to CAM than previously recognized.

[1] Plant Biology Section, School of Integrative Plant Science, Cornell University, Ithaca, NY, USA. [2] Boyce Thompson Institute, Ithaca, NY, USA. [3] Institute of Molecular & Cellular Biology, National Tsing Hua University, Hsinchu, Taiwan. [4] Department of Applied Bioinformatics, Institute for Microbiology and Genetics, University of Goettingen, Goettingen, Germany. [5] Campus Institute Data Science, University of Goettingen, Goettingen, Germany. [6] Department of Applied Bioinformatics, Goettingen Center for Molecular Biosciences, University of Goettingen, Goettingen, Germany. [7] Taiwan Forestry Research Institute, Taipei, Taiwan. [8] Department of Integrative Biology, University of Texas at Austin, Austin, TX, USA. [9] Department of Ecology and Evolutionary Biology, University of Arizona, Tucson, AZ, USA. [10] The Molecular and Cellular Biology Laboratory, The Salk Institute for Biological Studies, La Jolla, CA, USA.
✉email: tmichael@salk.edu; fl329@cornell.edu

*Isoetes*, commonly known as quillworts, is the only genus in the lycophyte order Isoetales, containing roughly 250 described species[1]. It is the last remaining member of an ancient lineage with a fossil record that dates back to at least the late Devonian. As such, quillworts are believed to represent the closest living relatives of the giant, tree-like lycopsids such as *Sigillaria* and *Lepidodendron* that dominated the terrestrial landscape during the Carboniferous[2]. However, in contrast to its arborescent ancestors, modern *Isoetes* species are diminutive and mostly aquatic with the vast majority of species growing completely or partially submerged. Underwater, *Isoetes* can conduct CAM[3], a carbon concentrating mechanism involving the separation of carbon uptake and fixation in a time of day (TOD) fashion, with carbon being sequestered as malate at night, to be fed into the Calvin cycle during the day. CAM is a common strategy to improve water-use efficiency among xeric-adapted plants, allowing them to keep their stomata closed during the day. However, its prevalence in aquatic species of *Isoetes*[3], as well as several aquatic angiosperms[4,5], highlights its utility for reducing photorespiration where $CO_2$ availability may be limited. While $CO_2$ limitation in terrestrial plants is caused by increased stomatal resistance, in aquatics it is largely the result of the relatively high diffusional resistance of water combined with significant diel fluctuation of dissolved $CO_2$ in the oligotrophic lakes and seasonal pools[4,6].

Though it has been nearly four decades since Keeley first described "CAM-like diurnal acid metabolism" in *Isoetes howellii*[7], relatively little is known about the genetic mechanisms controlling CAM in *Isoetes* or any other aquatic plant. Previous genomic and/or transcriptomic studies that focused on terrestrial CAM have found evidence for regulatory neofunctionalization, enrichment of *cis*-regulatory elements, and/or reprogramming of gene regulatory networks that underlie the convergent evolution of CAM in *Sedum album*[8], *Ananas comosus*[9], *Kalanchoe fedtschenkoi*[10], several orchids[11–13], and Agavoideae species[14,15]. Furthermore, a case of amino acid sequence convergence in phosphoenolpyruvate carboxylase (PEPC), which catalyzes the carboxylation of phosphoenolpyruvate (PEP) to yield oxaloacetate (OAA), has also been reported among some terrestrial CAM plants. However, the lack of a high-quality genome assembly has made a meaningful comparison of *Isoetes* or any other aquatic CAM plant to terrestrial CAM species impossible.

The only lycophyte genomes available to date are from the genus *Selaginella*[16–18], leaving a deep, >300-million-year gap in our knowledge of lycophyte genomics and limiting inferences of tracheophyte evolution. *Selaginella* is the only genus in the Selaginellales, the sister clade to Isoetales. Notably, *Selaginella* is known for being one of few lineages of vascular plants for which no ancient whole-genome duplications (WGDs) have been detected. Conversely, there is evidence from transcriptomic data for as many as two rounds of WGD in *Isoetes tegetiformans*[19]. As such, a thorough characterization of the history of WGD in *Isoetes* is vital to future research into the effects and significance of WGD across lycophyte diversity.

With this study, we seek to investigate genome evolution as well as the genetic underpinnings of CAM in *Isoetes*. To that end, we present a high-quality genome assembly for *Isoetes taiwanensis* DeVol. We find evidence for a single ancient WGD event that appears to be shared among multiple species of *Isoetes*. Additionally, while many CAM pathway genes display similar expression patterns in *Isoetes* and terrestrial angiosperms, notable differences in gene expression suggest that the evolution of CAM in *Isoetes* may have followed a markedly different path than it has in terrestrial angiosperms.

## Results

**Genome assembly, annotation, and organization.** Using Illumina short-reads, Nanopore long-reads, and Bionano optical mapping, 90.13% of the diploid ($2n = 2X = 22$ chromosomes) *I. taiwanensis* genome was assembled into 204 scaffolds (N50 = 17.40 Mb), with the remaining 9.87% into 909 unplaced contigs (Table 1). The total assembled genome size (1.66 Gb) is congruent with what was estimated by k-mers (1.65 Gb) and flow cytometry (1.55 Gb) (Supplementary Fig. 1). A circular-mapping plastome was also assembled, from which we identified a high level of RNA-editing (Supplementary Note 1 and Supplementary Fig. 2).

A total of 39,461 high confidence genes were annotated based on ab initio prediction, protein homology, and transcript evidence. The genome and proteome BUSCO scores are 94.5% and 91.0%, respectively, which are comparable to many other seed-free plant genomes (Supplementary Fig. 3) and indicative of high completeness. Orthofinder[20] analysis of 25 genomes placed 647,535 genes into 40,144 orthogroups (Supplementary Note 2 and Supplementary Fig. 4). Subsequent examination of lignin biosynthesis genes in *I. taiwanensis* suggests that evolution of particular pathway steps to S-lignin likely predates the divergence of *Isoetes* and *Selaginella* (Supplementary Note 3 and Supplementary Figs. 5-17). In addition, analysis of key stomatal and root genes (Supplementary Notes 4 and 6) in *I. taiwanensis* genome supported their homology (at the molecular level) with similar structures in other vascular plants (Supplementary Table 1 and Supplementary Figs. 18–20).

Repetitive sequences accounted for 38% of the genome assembly with transposable elements (TEs) accounting for the majority of those at 37.08% of the assembly length. Long terminal repeat (LTR) retrotransposons were the most abundant (15.72% of total genome assembly) with the Gypsy superfamily accounting for around 68% of LTR coverage (10.7% of total genome assembly; Supplementary Data 1). When repeat density was plotted alongside gene density, the distribution of both was found to be homogeneous throughout the assembly (Fig. 1). This even distribution of genes and repeats is markedly different from what has been reported in most angiosperm genomes[21] where gene density increases near the ends of individual chromosomes. However, it is consistent with several high-quality genomes published from seed-free plants, including *Physcomitrium patens*[22], *Marchantia polymorpha*[23], and *Anthoceros agrestis*[24]. The result from *I. taiwanensis* thus adds to the growing evidence that the genomic organization might be quite different between seed and seed-free plants[25].

**Table 1 *Isoetes taiwanensis* genome assembly statistics.**

| | |
|---|---|
| Assembly size (Mb) | 1658.30 |
| Scaffolds (#) | 204 |
| Scaffold length (Mb) | 1494.58 |
| N50 of scaffold length (Mb) | 17.40 |
| Scaffolded contigs (#) | 1879 |
| Scaffolded contig length (Mb) | 1211.25 |
| N50 length of scaffolded contigs (Mb) | 1.48 |
| Unscaffolded contig (#) | 909 |
| Unscaffolded contig (Mb) | 149.46 |
| N50 length of unscaffolded contigs (Mb) | 0.26 |
| Genome BUSCO score (Eukaryota) (%) | 94.5 |
| Proteome BUSCO score (Eukaryota) (%) | 91.0 |
| Predicted protein-coding genes (#) | 39,461 |
| Predicted repetitive sequence (%) | 38 |

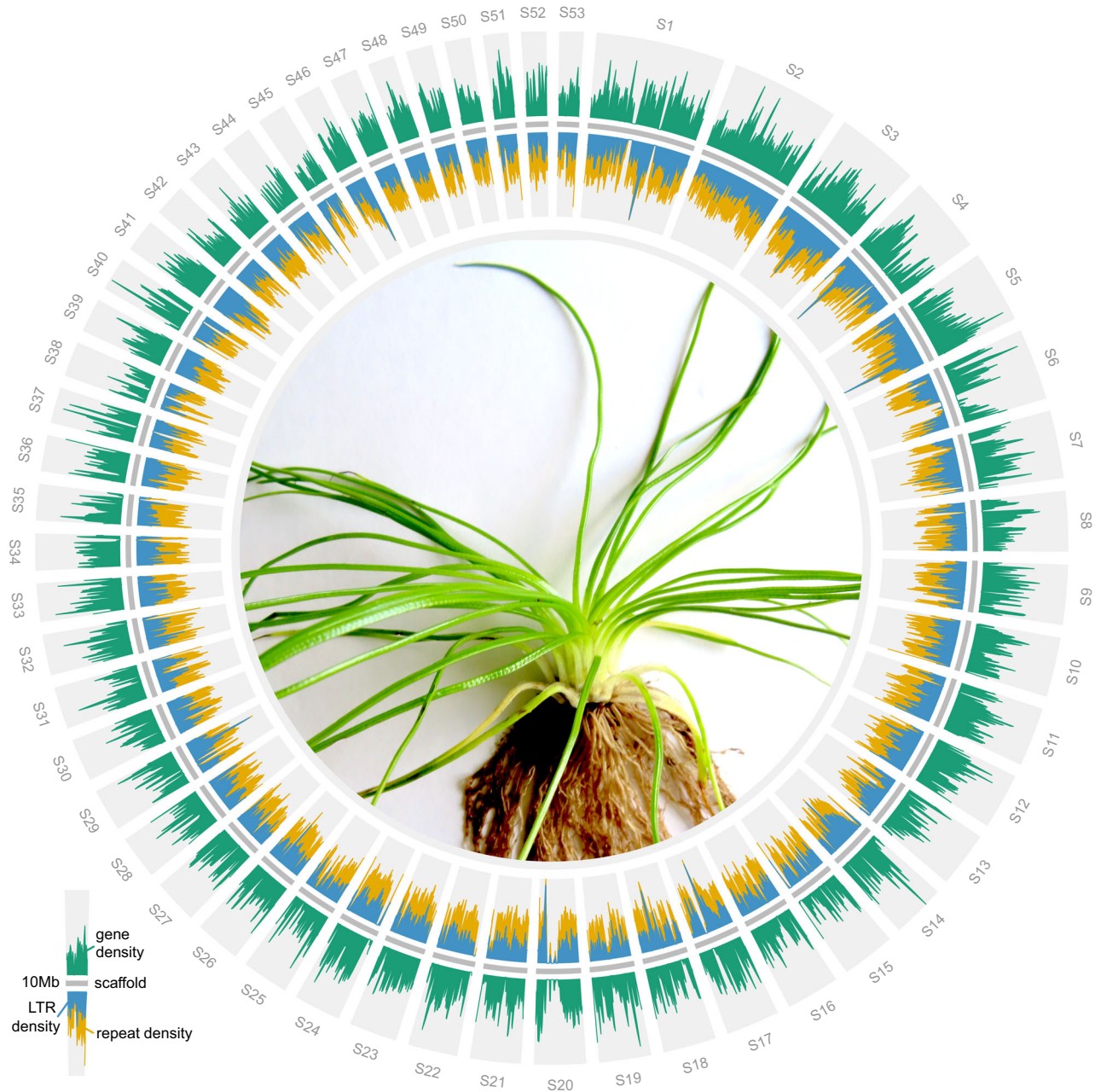

**Fig. 1 Distribution of genes and repetitive elements in *I. taiwanensis*.** The relatively even distributions differ from angiosperm genomes, but are similar to what have been reported in other seed-free plants. Only scaffolds longer than 10 Mb are plotted. Center: an image of *I. taiwanensis*. Source data are provided as a Source Data file.

**Evidence for WGD in *Isoetes taiwanensis*.** Using a combination of methods including synonymous substitutions per site ($Ks$), phylogenetic reconciliation, and synteny analyses, we identified a single ancient WGD in *I. taiwanensis*. This is in contrast to a previous $Ks$ analysis using 1KP transcriptome data, which found evidence for two rounds of WGD, named ISTEα and ISTEβ, in the North American species *I. tegetiformans* and *I. echinospora*[26]. These two WGDs have median $Ks$ values of ~0.5 and ~1.5[26] (Supplementary Fig. 21). Our $Ks$ analysis of the whole paranome (i.e., all of the paralogous gene copies in the genome) in *I. taiwanensis* revealed a single peak at $Ks$ ~ 1.8 (Fig. 2a), suggesting that the earlier of the two duplications (ISTEβ) in *I. tegetiformans* and *I. echinospora* is shared by *I. taiwanensis* while the more recent event (ISTEα) is not. This result was corroborated using

four-fold degenerate site transversion rates (4dtv; Supplementary Fig. 21). Further analysis of orthologous divergence between *I. taiwanensis* and *I. lacustris* indicated that ISTEβ predates the divergence of these two species (Supplementary Fig. 22). The ISTEβ event was subsequently confirmed by gene tree-species tree reconciliation using genomic data in the WhALE package[27]. WhALE returned a posterior distribution of gene retention centered on $q$ = ~0.12. This result compares favorably with a previously documented WGD event in *Azolla filiculoides*[28] ($q$ = ~0.08) and is in stark contrast to our negative control, *Marchantia polymorpha*[23] ($q$ = ~0) (Fig. 2b, c).

While self-self syntenic analysis revealed 6196 genes (15.7%) with a syntenic depth of 1× in 107 clusters (Supplementary Fig. 23), we do not believe they resulted from WGD. Our $Ks$

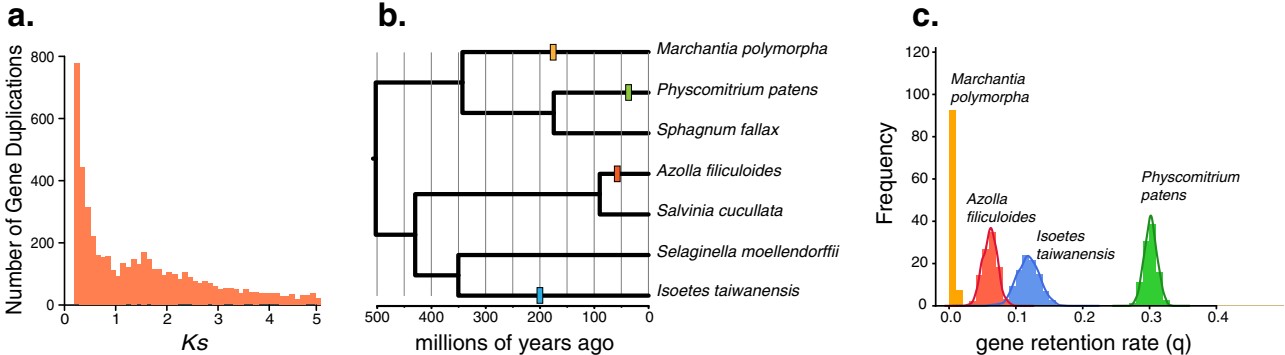

**Fig. 2 Evidence for WGD in *I. taiwanensis*. a** *K*s plot showing a peak centered on 1.8 corresponding to the ISTEβ event. **b** Hypothesized WGD events that were tested (colored rectangles) in our WhALE analysis are shown on a phylogeny. **c** *I. taiwanensis'* posterior distribution of gene retention rates (*q*) falls between that of *A. filiculoides* and *P. patens*, both are known to have at least one WGD. This provides additional support for the ISTEβ event. Conversely, the gene retention rate is close to zero for *M. polymorpha*, consistent with its lack of WGD. Source data are provided as a Source Data file.

analysis restricted to syntenic gene pairs failed to recover the peak at *K*s ~ 1.8 and instead consisted of an initial slope toward a much lower *K*s value (Supplementary Fig. 24). Given their high degree of similarity and location on separate scaffolds, it is possible that these low *K*s gene pairs are the result of relatively recent segmental duplications. The absence of conserved synteny from ISTEβ is unsurprising. The high *K*s value implies that ISTEβ is ancient; long enough ago for extensive genomic restructuring and fractionation to have taken place. Altogether, of the two hypothesized WGDs in *Isoetes*, we confirmed the presence of ISTEβ while the younger ISTEα might be either specific to *I. tegetiformans* and *I. echinospora* or an artifact stemming from the quality or completeness of the transcriptomes.

**Similarities to terrestrial CAM plants.** The CAM in *Isoetes* is unusual for at least two reasons. First, *Isoetes* diverged from other CAM plants more than 300 million years ago and second, *Isoetes* has an aquatic lifestyle. Here, we demonstrated that when submerged, titratable acidity in the leaves of *I. taiwanensis* increased throughout the night, reaching peak acidity in the morning and decreased throughout the daylight hours (Fig. 3b), consistent with the cycle of carbon sequestration and assimilation seen in dry-adapted CAM plants. To identify the underlying genetic elements, we generated TOD RNA-seq (Supplementary Data 2), sampling every 3 h over a 27 h period under 12 h light/12 h dark and continuous temperature (LDHH). A multidimensional scaling (MDS) plot of normalized expression data showed that the samples were generally clustered in a clockwise fashion as expected for TOD expression analysis (Supplementary Fig. 25).

We found that some of the CAM pathway genes in *I. taiwanensis* exhibited TOD expression patterns that largely resemble those found in terrestrial CAM plants (Fig. 3c–i and Supplementary Data 3). For example, the strong dark expression of *PHOSPHOENOLPYRUVATE CARBOXYLASE KINASE* (*PPCK*) appears to be conserved in *I. taiwanensis* as well as in all three terrestrial taxa (Fig. 3i). Likewise, we found one copy of *β-CARBONIC ANHYDRASE* (*β-CA*) that cycled similarly with homologs in *A. comosus* and *K. fedtschenkoi* (Fig. 3g)—increasing during the night and peaking in the early morning—although this is different from *S. album* in which no *β-CA* genes showed a high dark expression. Similar to *A. comosus* where two copies of *MALATE DEHYDROGENASE* (*MDH*) were found to cycle in green leaf tissue[9], we found multiple copies of *MDH* that appear to cycle in *I. taiwanensis* with one copy appearing to exhibit a similar peak expression to its orthologue in pineapple (Fig. 3e).

However, neither of the other two *MDH* genes that cycle in *I. taiwanensis* exhibit similar expression to their orthologues in terrestrial CAM species (Supplementary Fig. 26).

During the day, decarboxylation typically occurs by one of two separate pathways (Fig. 3a). The first utilizes NAPD-MALIC ENZYME (NADP-ME) and PYRUVATE PHOSPHATE DIKI-NASE (PPDK), and appears to be favored by *K. fedtschenkoi* and *S. album*[8,10]. The second utilizes MDH and PHOSPHOENOL-PYRUVATE CARBOXYKINASE (PEPCK) and is favored by *A. comosus*[9]. Based on its TOD expression of multiple copies of *MDH* and associated expression dynamics, it is possible that *I. taiwanensis* utilizes the MDH/PEPCK pathway. While all four genes have elevated expression levels during the day, the expression of *NADP-ME* is inverted compared to *K. fedtschenkoi* and *S. album* (Fig. 3c), and *PPDK* exhibits relatively weak cycling overall (R = 0.637; Fig. 3d). Additionally, *PEPCK* and one copy of *MDH* have similar TOD expression in *I. taiwanensis* and *A. comosus* (Fig. 3f, e, respectively), which may indicate a shared affinity for *MDH/PEPCK* decarboxylation. Interestingly, the copy of *PEPCK* that cycles in *I. taiwanensis* is not orthologous to the copy that cycles in *A. comosus*, being placed in a different orthogroup by Orthofinder[20].

**I. taiwanensis likely recruited bacterial-type PEPC.** While TOD expression of many key CAM pathway genes was broadly similar to that seen in terrestrial CAM plants, one important difference can be found in the PEPC enzyme, which is the entry point of carboxylation in CAM and C4 photosynthesis (Fig. 3a). PEPC is present in all photosynthetic organisms as well as many non-photosynthetic bacteria and archaea. It is a vital component of plant metabolism, carboxylating PEP in the presence of $HCO_3^-$ to yield OAA. In plants, the *PEPC* gene family consists of two clades, the "plant-type" and the "bacterial-type". The latter was named because of its higher sequence similarity with proteo-bacteria *PEPC* than other plant-type *PEPC* genes[29]. All CAM and C4 plants characterized to date recruited only the plant-type *PEPC*[30], with the bacterial-type often being expressed at relatively low levels and/or primarily in non-photosynthetic tissues[31].

Interestingly, in *I. taiwanensis* we found that both types of *PEPC* were cycling and that the bacterial-type was expressed at much higher levels than plant-type *PEPC* (Fig. 3h). Copies from both types had similar expression profiles in *I. taiwanensis*, peaking at dusk and gradually tapering off during the night. While this may seem counterintuitive as PEPC is an important component of the dark reactions, it is consistent with what has

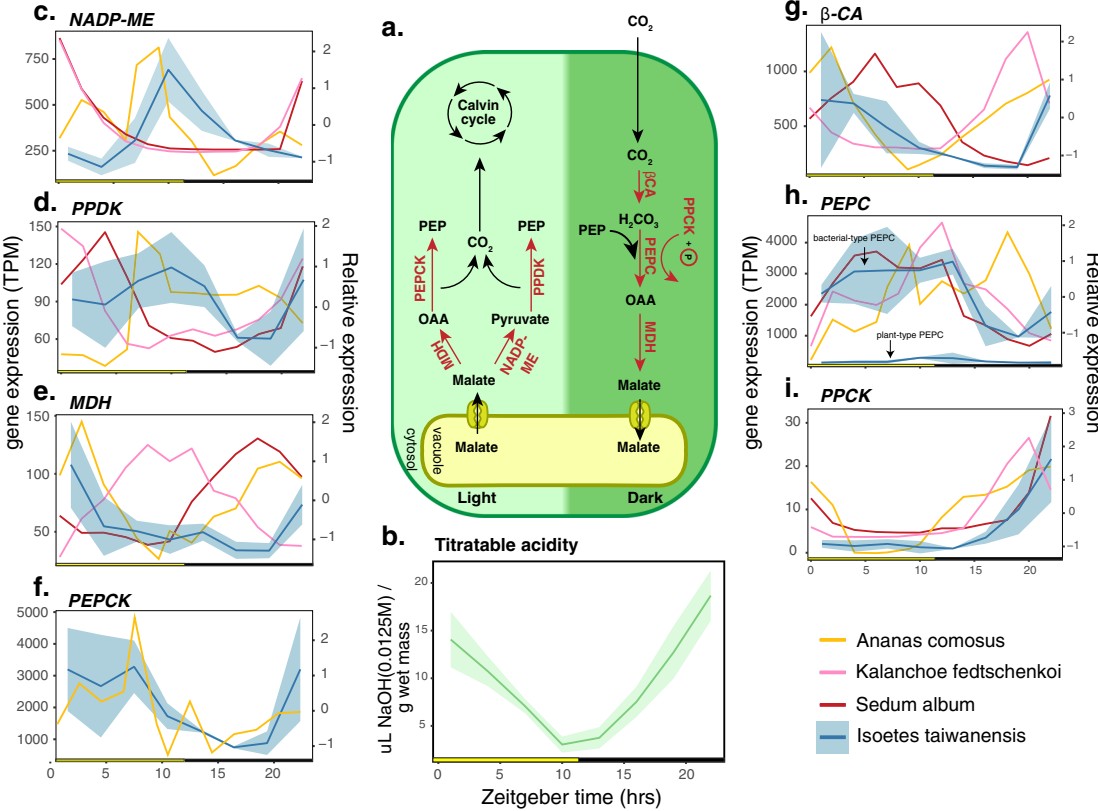

**Fig. 3 Key CAM pathway genes and their expression patterns in *I. taiwanensis*. a** The CAM pathway with important reactions and their enzymes shown in red. **b** Titratable acidity in *I. taiwanensis* exhibited a clear diel fluctuation. Diel expression patterns for highlighted genes are shown for the day (**c–f**) and night reactions (**g–i**). Average of TPM normalized expression data for *I. taiwanensis* is plotted in blue with a shaded ribbon representing the standard deviation. Relative expression profiles for homologous, cycling genes in other CAM species are plotted for comparison. All times are displayed in hours after lights-on (Zeitgeber time). Locus IDs for genes used in expression plots is provided in Supplementary Data 3. Source data for *I. taiwanensis* are provided in Supplementary Data 2. Source data for the other taxa are provided as a Source Data file.

previously been found in other terrestrial CAM plants, with the overall expression profile resembling that of *S. album*[8]. The advantage of recruiting bacterial-type *PEPC* is unclear. In vivo, both bacterial- and plant-type PEPC can interact with each other to form a hetero-octameric complex that is less sensitive to inhibition by malate[32]. Although the functional and physiological implications await future studies, the unusual involvement of bacterial-type PEPC is suggestive of a divergent evolutionary path to underwater CAM in *Isoetes*.

**No evidence for convergent evolution of PEPC**. Plant-type PEPC was recently shown to undergo convergent amino acid substitutions in concert with the evolution of CAM[10]. An aspartic acid (D) residue appears to have been repeatedly selected across multiple origins of CAM such as in *K. fedtschenkoi* and *P. equestris*[10], although notably not in *A. comosus* nor *S. album*. This residue is situated near the active site, and based on in vitro assays, the substitution to aspartic acid significantly increased PEPC activity[10]. However, in *I. taiwanensis* we did not observe the same substitution in any copies of PEPC (Fig. 4); instead, they have arginine (R) or lysine (H) at this position like PEPC from many non-CAM plants. This lack of sequence convergence between *Isoetes* and few CAM angiosperms could be the result of their substantial phylogenetic distance and highly divergent life histories. Alternatively, it is also likely that the substitution is not as important as previously hypothesized, or relevant only in the context of plant-type PEPC. As *I. taiwanensis* may also utilize

bacterial-type PEPC, the aspartic acid residue might not serve the same purpose.

**Circadian regulation in *Isoetes***. Previous analysis of the *A. comosus* genome found promoter regions of multiple key CAM pathway genes containing known circadian *cis*-regulatory elements (CREs) including Morning Element (ME: CCACAC), Evening Element (EE: AAATATCT), CCA1-binding site (CBS: AAAAATCT), G-box (CACGTG) and TCP15-binding motif (NGGNCCCAC)[9]. This suggests that expression of CAM genes in pineapple is largely under the control of a handful of known circadian clock elements. The direct involvement of circadian CREs was corroborated by a later study of the facultative CAM plant *S. album* where shifts in diel expression patterns were tied to a shift in TOD-specific enrichment of CREs: EE and Telobox (TBX: AAACCCT)[8].

In order to examine the role of the circadian clock and light/dark cycles in regulating *I. taiwanensis* CAM, we used the HAYSTACK pipeline[33] to identify all genes with TOD expression patterns. We predicted 3241 cycling genes, which is 10% of the expressed genes (Supplementary Fig. 27 and Supplementary Note 6). While 10% is low compared to land plants that have been tested under this condition (LDHH)—usually at 30–50% genes[8,33,34], a recent study found a reduced number of cycling genes in another aquatic plant *Wolffia australiana* (duckweed/watermeal)[35]. Accordingly, decreased cycling may be a feature of aquatic plants. Further discussions and comparisons of *I. taiwanensis* TOD gene expression with other species can be

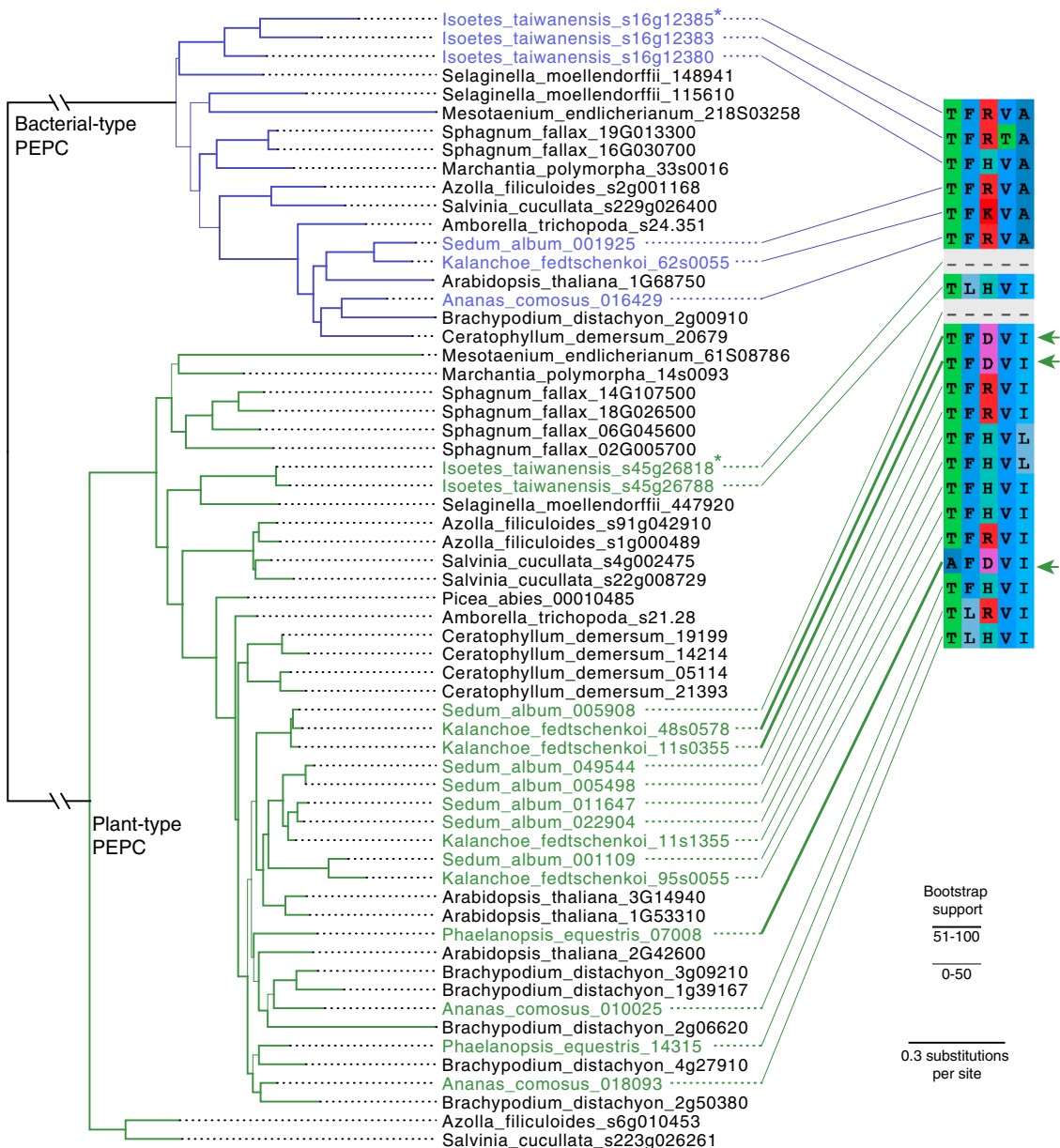

**Fig. 4 A lack of PEPC sequence convergence in *I. taiwanensis*.** Copies with putative convergent amino acid sequence (D at position 3 in alignment) are indicated by thickened connecting lines and green arrows. Copies of bacterial-type and plant-type PEPC shown to cycle in *I. taiwanensis* are marked with asterisks (*). Branch thickness indicates bootstrap support.

found in Supplementary Note 7, Supplementary Figs. 27, 28, and Supplementary Data 4.

Core circadian clock genes such as *LATE ELONGATED HYPOCOTYL* (*LHY*), *PSEUDO-RESPONSE REGULATOR 7* (*PRR7*), *LUX ARRHYTHMO* (*LUX*), and *EARLY FLOWERING 3* (*ELF3*), cycle with the expected TOD expression seen in their *Arabidopsis* orthologs (Fig. 5, Supplementary Note 8, and Supplementary Figs. 29, 30)[33]. Furthermore, *TIMING OF CAB2 1/PSEUDO-RESPONSE REGULATOR 1* (*TOC1/PRR1*) and *GIGANTEA* (*GI*), which are typically single-copy genes in land plants, have, respectively, 3 and 5 predicted genes in distinct genomic locations. Similarly, an increased number of homologs was found in the facultative CAM plant *S. album*[8]. Closer inspection confirmed all 3 *TOC1/PRR1* paralogs are full length, while only 1 of the *GI* genes (*GIa*) is full length and 1 other (*GIb*) is a true partial/truncated (and expressed) paralog. Surprisingly, all 3 copies of

*TOC1/PRR1* have dawn-specific expression compared to the dusk-specific expression found in all plants tested to date[36] (Fig. 5b) including terrestrial CAM species (Supplementary Fig. 30). In addition, *GIa* and *GIb* have antiphasic expression, with the full-length *GIa* having dusk-specific expression, which is consistent with other plants, and *GIb* having dawn-specific expression (Fig. 5c and Supplementary Fig. 30).

The duplications and divergent expression patterns of *TOC1/PRR1* and *GI* in *I. taiwanensis* have important implications on circadian clock evolution. Despite the TOD expression of core circadian clock genes being highly conserved since the common ancestor of green algae and angiosperms, the mechanisms may be simpler in algae[37] and mosses[38]. This idea is largely based on the lack of key components of the evening-phased loop including *PRR1*, *GI*, and *ZTL* in *P. patens* and the absence of the same along with morning-phased loop genes *ELF3* and *ELF4* in algae[39].

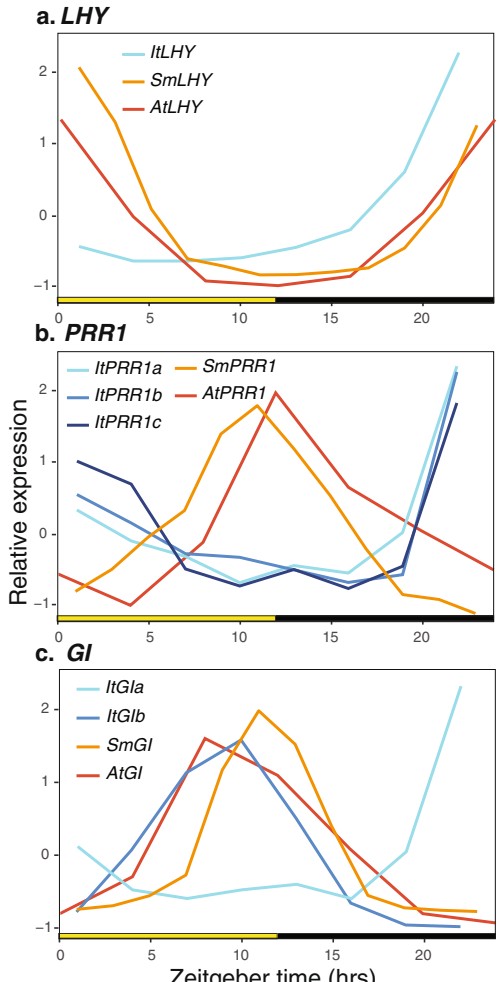

**Fig. 5 Expression of key circadian associated genes is shifted in _I. taiwanensis_. a** _LATE ELONGATED HYPOCOTYL (LHY)_, **b** _PSEUDO-RESPONSE REGULATOR 1 (PRR1)_, **c** _GIGANTEA (GI)_ orthologs in _Isoetes_ (blue lines), _Selaginella_ (orange line), and _Arabidopsis_ (red line) normalized expression over the day. Day (yellow box); night (black box); Zeitgeber time (ZT) is the number of hours (h) after lights on (0 h). Locus IDs for genes used in expression plots can be found in Supplementary Data 3. Source data for _I. taiwanensis_ are provided in Supplementary Data 2. Source data for the other taxa are provided as a Source Data file.

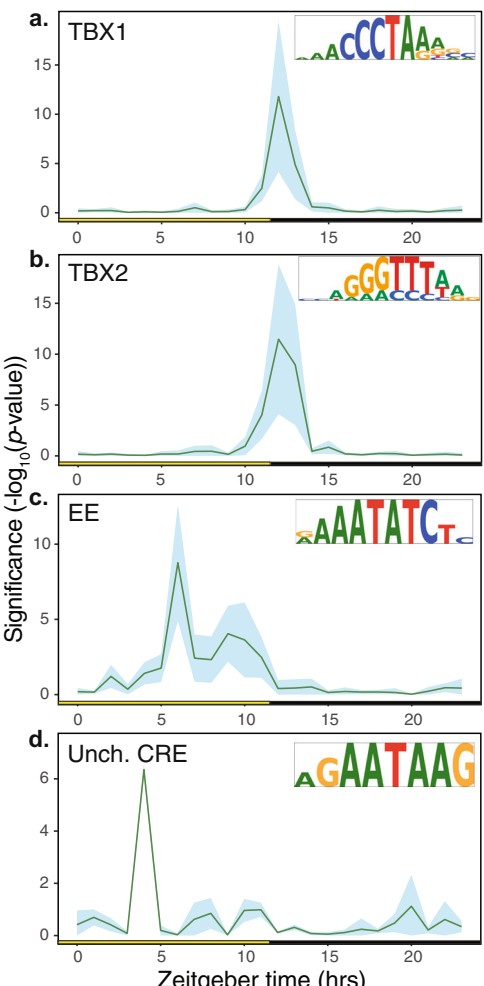

**Fig. 6 Multiple CREs exhibit time-structured enrichment in _I. taiwanensis_. a**, **b** Two telobox (TBX) containing motifs showed similar patterns to one another, both being enriched ingenes with peak expression at dusk. **c** Motif containing Evening Element (EE) was significantly enriched in genes with peak expression at mid-day. **d** A significantly enriched motif at mid-day. Day (yellow box); night (black box); Zeitgeber time (ZT) is the number of hours (h) after lights on (0 h). Shaded regions represent the standard deviation of log-transformed, FDR corrected, single-tailed _p_-values of component k-mers as calculated by ELEMENT[33]. Source data are provided as a Source Data file.

While _I. taiwanensis_ possesses all the major clock genes that are found in other vascular plants, lineage-specific expansion and phase-shifted gene expression in the evening-phased loop could indicate that circadian control was less conserved during the early evolution of land plants. However, _Selaginella_ exhibits a very similar expression of various circadian modules relative to other vascular plants and likewise, possesses a single copy of both _GI_ and _PRR1_[39]. It is thus possible that the TOD architecture in _I. taiwanensis_ represents a more recent adaptation to its aquatic CAM lifestyle. As a comparison, _S. album_ similarly has multiple duplicated clock genes and its transition to CAM is associated with significant shifts in both phase and amplitude of gene expression[8]. To further investigate the relationship between clock and CAM in _I. taiwanensis_, we next focused on characterizing the circadian CREs.

**Canonical circadian CREs are not enriched in _Isoetes_ CAM cycling genes.** We used ELEMENT[33] to exhaustively search the

promoter region of cycling genes for putative CRE motifs. Following de novo identification, putative CREs were compared to known transcription factor binding sites in _Arabidopsis_ to determine to what degree their functions might be conserved between _Isoetes_ and flowering plants. We identified 16 significantly enriched CREs motifs in the 500 bp 5′ promoter region of cycling genes identified by HAYSTACK, and clustered them according to TOD expression (Supplementary Data 5). Half of the motifs shared some degree of sequence similarity to known circadian CREs previously identified in _Arabidopsis_, including the EE as well as two "ACGT"-containing elements (G-box-like) and two TBX-containing motifs[33]. In the case of TBX, both motifs were associated with peak expression at dusk (at around 12 h after lights on; Zeitgeber Time [ZT]) in _I. taiwanensis_ (Fig. 6a, b), similar to _Arabidopsis_ under light/dark cycles alone[33]. On the other hand, the EE appear to be associated with peak expression at different TOD. In _Arabidopsis_, the EE is enriched in genes with peak expression at dusk (ZT = 12), but in _I. taiwanensis_, this

pattern is shifted, with the EE associating with genes that peak in expression around mid-day (ZT = 6) (Fig. 6c). Additionally, while the two "ACGT"-containing elements were found upstream of genes that exhibited significant cycling behavior, neither was strongly associated with peak expression at a particular TOD. We also found an unidentified CRE (AGAATAAG) that is strongly associated with peak expression in the morning (ZT = 4) (Fig. 6d).

We next examined the connection between circadian CREs and CAM genes in *I. taiwanensis*. Interestingly, with the exception of the RVE1/2 motif, we did not find significant enrichment of any known circadian CREs in CAM cycling genes relative to non-cycling paralogues. While a targeted search of CAM cycling gene promoters did uncover circadian CREs including the CBS, TCP15, TBX, and EE (Supplementary Data 6), none were strongly associated with either light or dark phase CAM gene expression. In addition, both ME and G-box were conspicuously absent from the promoter regions of cycling CAM photosynthetic genes.

In sum, TOD-specific enrichment of CREs appears to differ in various aspects from *Arabidopsis*. While some CRE sequences themselves are conserved between lycophytes and angiosperms, their interaction with various transcription factors and subsequent regulatory function could be quite different in *Isoetes*. Importantly, our results stand in contrast to other CAM plants such as *S. album*[8] and *A. comosus*[9] where CAM genes appeared to be under the direct control of a handful of strictly conserved circadian CREs. These results either suggest that the circadian clock network that emerged in *Isoetes*, which included the addition of central components *GI* and *PRR1*, was quite different than that found to be highly conserved in seed plants, or there is significant TOD innovation associated with the evolution of underwater CAM. Additional *Isoetes* genomes and TOD analysis of underwater CAM plants will be required to test these hypotheses.

The assembly and analyses of the *I. taiwanensis* genome bridges a substantial gap in our knowledge of vascular plant evolution. We have combined genomic and transcriptomic data to corroborate one of the two hypothesized WGDs in *Isoetes* relative to its closest extant relative *Selaginella*, highlighting the contrasting history of WGD in these two lineages. Additionally, comparison of TOD gene expression with genomic sequence data has given us insights into the convergent evolution of CAM photosynthesis, not only in a lycophyte, but also in the aquatic environment. As such, our analysis stands as a necessary counterpoint to similar studies previously conducted in terrestrial angiosperms. Shifts in expression of CAM pathway genes and the possible recruitment of bacterial-type PEPC in *I. taiwanensis* demonstrate a remarkable degree of plasticity in the convergent evolution of this complex trait throughout vascular plants. Likewise, differences in the enrichment of CREs associated with circadian gene expression suggest that control of CAM, as well as other processes tied to the circadian clock, may have diverged markedly since the common ancestor of *Isoetes* and flowering plants. We propose that the emergence of underwater CAM may have followed a distinct route in *Isoetes*, shedding light on a classic example of convergent evolution of a complex plant trait.

## Methods

**Plant sample.** *Isoetes taiwanensis* is endemic to a small pond in Northern Taiwan and has been ex situ propagated in Taiwan Forestry Research Institute. This species is expected to have a low genetic diversity due to a very restricted distribution and a small population size. The voucher specimen (Kuo4500) was deposited at TAIF herbarium.

**Genome size estimate.** The genome size of *I. taiwanensis* was first determined by flow cytometry following the protocols outlined in Kuo et al.[40] and Li et al.[28]. The

flow cytometric experiments were performed on BD FACSCan system (BD Biosciences, USA), and the Beckman buffer[41] was used with 0.5% (v/v) 2-mercaptoethanol, 40 mg mL$^{-1}$ PVP-40, and 0.1 mg mL$^{-1}$ RNaseA added. We used *Zea mays* (1C = 5.57pg[42]) as the internal standard. To confirm the flow cytometry-based measurement, a k-mer frequency distribution was generated from Illumina 2 × 150 bp paired reads (described below) using Jellyfish[43], which was then input into GenomeScope[44] to estimate genome size.

**Genome sequencing.** High molecular weight (HMW) DNA was extracted using a modified CTAB method on isolated nuclei. First, leaf tissues were ground in liquid nitrogen, and the powder was resuspended in the Beckman buffer (same as in our flow cytometric experiments). We then used 30 μm nylon circular filters (Partec, Germany) to remove tissue debris, and precipitated nuclei with 100×*g* centrifugation under 4 °C for 20 min. DNA was extracted following a modified CTAB protocol[45]. HMW DNA was QC'd on an agarose gel for length and quantified on a bioanalyzer. Unsheared HMW DNA was used to make Oxford Nanopore Technologies (ONT) ligation-based libraries (Oxford, UK). Libraries were prepared starting with 1.5 μg of DNA and following all other steps in ONT's SQK-LSK109 protocol. Final libraries were loaded on an ONT flowcell (v9.4.1) and run on the GridION. Bases were called in real-time on the GridION using the flip-flop version of Guppy (v3.1). The resulting fastq files were concatenated and used for downstream genome assembly steps. The same batch of HMW genomic DNA was used to construct Illumina (Illumina, USA) libraries for estimating genome size (above) and correcting residual errors in the ONT assembly. Libraries were constructed using the KAPA HyperPrep Kit (Kapa Biosystems, Switzerland) followed by sequencing on an Illumina NovaSeq6000 with 2 × 150 bp paired-ends.

**Genome assembly.** ONT reads were assembled using minimap2 and miniasm[46], and the resulting draft assembly was then polished by racon[47] (with ONT reads) and pilon[48] (with Illumina reads). Because the plants were grown non-axenically under water, the assembly inevitably contained contaminations. We, therefore, used blobtools[49] to identify non-plant contigs based on a combination of contig read coverage, taxonomic assignment, and GC content.

To further scaffold the assembly, we generated a genome map using Bionano with the Direct Label and Stain chemistry and DLE-1 labeling. For this, high molecular weight DNA was extracted using the Bionano Plant DNA Isolation Kit. Hybrid scaffolding, combining the nanopore draft and Bionano map, was done on the Bionano Saphyr computing platform at the McDonnell Genome Institute at Washington University. We then gap-filled the scaffolded genome using two rounds of LR_Gapcloser[50] (3 iterations each and a pilon polishing in between). Finally, to remove redundancy the purge_haplotigs pipeline[51] was used to obtain the v1 assembly. The circular chloroplast genome was assembled from Illumina data using the GetOrganelle[52] toolkit.

**Repeat annotation.** We generated a custom *I. taiwanensis*-specific repeat library using LTR-retriever[53] and RepeatModeler[54]. The *I. taiwanensis* genome contains a high number of genes encoding pentatricopeptide repeat proteins which are often misclassified as repetitive elements in the genome. Thus, in order to identify and remove repeats with homology to plant proteins, we used BLASTx to query each repeat against the uniprot plant protein database (*e*-value threshold at 1e−10). The resulting library was then input into RepeatMasker[55] to annotate and mask the repetitive elements in the *I. taiwanensis* genome.

**Gene annotation.** We trained two ab initio gene predictors, AUGUSTUS[56] and SNAP[57], on the repeat-masked genome using a combination of protein and transcript evidence. For the protein evidence, we relied on the annotated proteomes from *Selaginella moellendorffii*[16] and *S. lepidophylla*[18], and for the transcript evidence, we used the RNA-seq data from our time-course experiment and a separate corm sample. To train AUGUSTUS, BRAKER2[58] was used and the transcript evidence was input as an aligned bam file. SNAP was trained under MAKER with 3 iterations, and in this case, the transcript evidence was supplied as a de novo assembled transcriptome done by Trinity[59]. After AUGUSTUS and SNAP were trained, they were fed into MAKER[60] along with all the evidence to provide a synthesized gene prediction. Gene functional annotation was done using the eggNOG-mapper v2[61]. To filter out spurious gene models, we removed genes that met none of the following criteria: (1) a transcript abundance greater than zero in any sample (as estimated by Stringtie[62]), (2) has functional annotation from egg-NOG, and (3) was assigned into orthogroups in an Orthofinder[20] run (see below). The resulting gene set was used in all subsequent analyses.

**Homology assessment and gene family analysis.** Homology was initially assessed with Orthofinder[20] using genomic data from a range of taxa from across the plant tree of life including all CAM plant genomes published to date: *Amborella trichopoda*[63], *Ananas comosus*[9], *Anthoceros agrestis*[24], *Arabidopsis thaliana*[64], *Azolla filiculoides*[28], *Brachypodium distachyon*[65], *Ceratophyllum demersum*[66], *Isoetes taiwanensis* (this study), *Kalanchoe fedtschenkoi*[10], *Marchantia polymorpha*[23], *Medicago truncatula*[67], *Nelumbo nucifera*[68], *Nymphaea colorata*[69], *Phalaenopsis equestris*[11], *Physcomitrium patens*[22], *Picea abies*[70], *Salvinia cucullata*[28], *Sedum album*[8], *Selaginella moellendorffii*[16], *Sphagnum fallax* (*Sphagnum fallax* v0.5, DOE-

JGI, http://phytozome.jgi.doe.gov/), *Spirodela polyrhiza*[71], *Utricularia gibba*[72], *Vitus vinifera*[73], and *Zostera marina*[74], and one algal genome: *Mesotaenium endlicherianum*[75]. Following homology assessment, the degree of overlap between gene families was assessed using the UpsetR[76] package in R.

**RNA-editing analysis**. RNA-seq data were first mapped to combined nuclear and chloroplast genome assemblies using HISAT2[77]. The reads mapping to the chloroplast genome were extracted using samtools[78]. SNPs were called using the mpileup function in bcftools[79]. The resulting vcf files were filtered using bcftools to remove samples with a depth < 20, quality score < 20 and mapping quality bias < 0.05. After filtering, C-to-U and U-to-C edits were identified using an alternate allele frequency threshold of 10%. Finally, RNA-editing sites were related to specific genes using the intersect command in bedtools[80] and characterized using a custom python script (available at https://github.com/dawickell/Isoetes_CAM).

**Ks analysis**. Ks divergence was calculated by several different methods. Initially, a whole paranome Ks distribution was generated using the "wgd mcl" tool[81]. Self-synteny was then assessed in i-Adhore and Ks values were calculated and plotted for syntenic pairs only using the "wgd syn" tool[81]. To conduct Ks analysis of related species, RNA-seq data was downloaded from the SRA database for *Isoetes yunguiensis* (SRR6920723)[82], *I. sinensis* (SRR1648119)[83], *I. drummondii* (SRR4762161), *I. echinospora* (SRR6853338)[84], *I. lacustris* (SRR9620527)[85], and *I. tegetiformans* (ERR2040873)[19]. Transcriptomes were assembled using SOAPdenovo-Trans[86] with a k-mer length of 31. Next, for each *Isoetes* genome and transcriptome, we used the DupPipe pipeline to construct gene families and estimate the age distribution of gene duplications[87,88]. We translated DNA sequences and identified ORFs by comparing the Genewise[89] alignment to the best-hit protein from a collection of proteins from 25 plant genomes from Phytozome[90]. For all DupPipe runs, we used protein-guided DNA alignments to align our nucleic acid sequences while maintaining the ORFs. We estimated Ks divergence using PAML[91] with the F3X4 model for each node in the gene family phylogenies.

**Four-fold transversion substitution rate analysis**. For each *Isoetes* genome and transcriptome, we used the DupPipe pipeline as described above to generate gene alignments. We estimated a four-fold transversion substitution rate (4dtv) using an existing perl script for each duplicate gene pair (https://github.com/chaimol/KK4D/blob/master/calculate_4DTV_correction.pl).

**Estimation of orthologous divergence**. To place putative WGDs in relation to lineage divergence, we estimated the synonymous divergence of orthologs among pairs of species that may share a WGD based on their phylogenetic position and evidence from the within-species Ks plots. We used the RBH Orthologue pipeline[88] to estimate the mean and median synonymous divergence of orthologs, and compared those with the synonymous divergence of inferred paleopolyploid peaks. We identified orthologs as reciprocal best BLAST hits in pairs of transcriptomes. Using protein-guided DNA alignments, we estimated the pairwise synonymous divergence for each pair of orthologs using PAML[91] with the F3X4 model.

**Phylogenetic assessment of ancient whole-genome duplication**. WGD inference was conducted by phylogenomic reconciliation using the WhALE package implemented in Julia[27]. First, prior to WhALE analysis, Orthofinder[20] was used to identify groups of orthologous genes among 7 species representing 3 taxonomic groups (bryophytes, lycophytes, and ferns): *Azolla filiculoides*[28], *Isoetes taiwanensis* (this study), *Marchantia polymorpha*[23], *Physcomitrium patens*[22], *Salvinia cucullata*[28], *Selaginella moellendorffii*[16], and *Sphagnum fallax* (*Sphagnum fallax v0.5*, DOE-JGI, http://phytozome.jgi.doe.gov/). These species were chosen based on phylogenetic relatedness, availability of a high-quality genome assembly, and previous assessment for the presence or absence of WGD. The resulting orthogroups were filtered using a custom python script to remove the 5% largest orthogroups and those with less than 3 taxa. Additionally, WhALE requires removal of gene families that do not contain at least one gene in both bryophytes and ferns to prevent the inclusion of gene families originating after divergence from the most recent common ancestor. Alignments were generated for the filtered orthogroups in PRANK[92] using the default settings. A posterior distribution of trees was obtained for each gene family in MrBayes 3.2.6[93] using the LG model. Chains were sampled every 10 generations for 100,000 generations with a relative burn-in of 25%. Following the Bayesian analysis, conditional clade distributions (CCDs) were determined from posterior distribution samples using ALEobserve in the ALE software suite[94]. CCD files were subsequently filtered using the ccddata.py and ccdfilter.py scripts provided with the WhALE program. A dated, ultrametric species tree was generated using the "ape" package in R[95], in which branch lengths were constrained according to 95% highest posterior density of ages, assuming that bryophytes are monophyletic, as reported by Morris et al.[96]. Finally, the filtered CCD files were loaded in Julia along with the associated species phylogeny. A hypothetical WGD node was inferred at 200 million years ago (MYA) along the branch leading to *I. taiwanensis*, prior to the estimated crown age of extant *Isoetes*[97]. Modifying the hypothetical age of this WGD node did not affect the outcome. Additional WGD nodes were placed as positive controls along branches leading to *Physcomitrium patens* and *Azolla filiculoides* at 40 MYA and 60 MYA,

respectively, based on previous studies[22,28]. A false WGD event was also placed arbitrarily in *Marchantia polymorpha* at 160 MYA as a negative control. A WhALE "problem" was constructed using an independent rate prior and MCMC analysis was conducted using the DynamicHMC library in Julia (https://github.com/tpapp/DynamicHMC.jl) with a sample size of 1000.

**Phylogenetic analysis of root, stomata, and CAM pathway genes**. Following clustering of homologs in Orthofinder, we conducted a phylogenetic analysis of several gene families of interest, including those containing *SMF*, *FAMA*, *TMM*, *RSL*, and *PEPC* genes, which were identified based on homology using gene annotations from *Arabidopsis*. Gene trees from Orthofinder were initially used to identify paralogues and remove fragmented genes where appropriate. In the case of *PEPC*, orthogroups containing "bacterial-type" and "plant-type" *PEPC* were combined prior to alignment. Next, amino acid sequences were aligned using MUSCLE[98] under default settings and trimmed using TrimAL with the -strict flag. An amino acid substitution model was selected according to the Bayesian Information Criterion (BIC) in ModelFinder[99] prior to phylogenetic reconstruction by maximum likelihood in IQ-TREE v1.6.12[100] with 1000 ultrafast[101] bootstrap replicates.

**Phylogenetic and gene expression analysis of genes salient to the phenylpropanoid and lignin biosynthesis pathway**. The datasets used for phylogenetic analyses were based on de Vries et al.[102] with added *I. taiwanensis* sequences. In brief, we assembled a dataset of predicted proteins from (A) the genomes of seventeen land plants: *Anthoceros agrestis* as well as *Anthoceros punctatus*[24], *Amborella trichopoda*[63], *Arabidopsis thaliana*[64], *Azolla filiculoides*[28], *Brachypodium distachyon*[65], *Capsella grandiflora*[103], *Gnetum montanum*[104], *Isoetes taiwanensis* (this study), *Marchantia polymorpha*[23], *Nicotiana tabacum*[105], *Oryza sativa*[106], *Physcomitrium patens*[22], *Picea abies*[70], *Salvinia cucullata*[28], *Selaginella moellendorffii*[16], and *Theobroma cacao*;[107] (B) the genomes of seven streptophyte algae: *Chlorokybus atmophyticus*[108], *Chara braunii*[109], *Klebsormidium nitens*[110], *Mesotaenium endlicherianum*[75], *Mesostigma viride*[108], *Penium margaritaceum*[111], *Spirogloea muscicola*[75]—additionally, we included sequences found in the transcriptomes of *Spirogyra pratensis*[112], *Coleochaete scutata* as well as *Zygnema circumcarinatum*[113], and *Coleochaete orbicularis*;[114] (C) the genomes of eight chlorophytes: *Bathycoccus prasinos*[115], *Chlamydomonas reinhardtii*[116], *Coccomyxa subellipsoidea*[117], *Micromonas* sp. as well as *Micromonas pusilla*[118], *Ostreococcus lucimarinus*[119], *Ulva mutabilis*[120], *Volvox carteri*[121]. For phenylalanine ammonia-lyase, additional informative sequences were added based on de Vries et al.[122].

Building on the alignments published in de Vries et al.[102], homologs of each gene family (detected in the aforementioned species via BLASTp) were (re-)aligned using MAFFT v7.475[123] with a L-INS-I approach; both full and partial sequences from *I. taiwanensis* were retained. We constructed maximum likelihood phylogenies using IQ-TREE 2.0.6[124]; 1000 ultrafast[101] bootstrap replicates were computed. To determine the best model for protein evolution, we used ModelFinder[99] and picked the best models based on BIC (PAL: LG + F + R7; CSE: LG + F + R8; C4H: LG + R8; C3H: LG + R10; COMT: JTT + R7; HCT: WAG + R9; F5H: LG + F + R10; CCoAOMT: WAG + R5; 4CL: LG + R9; CAD: LG + R8; CCR: LG + R6). Residue information was mapped next to the tree based on structural analyses by Hu et al.[125], Pan et al.[126], Louie et al.[127], Youn et al.[128], and Ferrer et al.[129].

Raw read counts from expression data of different tissues (leaf time series and the corm) of *I. taiwanensis* were extracted for those genes highlighted. Data were filtered to retain genes with more than 1 count per million (CPM) in at least 2 samples and normalized by applying the trimmed mean of M values procedure with the edgeR package[130]. The count data were transformed to log2-CPM via the limma package[131]. Finally, heatmaps of gene expression levels were produced via the R packages gplots and RColorBrewer.

**Time-course titratable acidity and RNA-seq experiments**. Leaves of *I. taiwanensis* were taken from five individuals (as five biological replicates) every 3 h over a 27-h period on a 12-hour light/dark cycle and constant temperature. To measure changes in acidity over time, a portion of the leaf tissues was weighed, mixed with 3.5–5 mL of ddH$_2$O, and titrated with 0.0125 M NaOH solution until pH = 7.0. At the same time, we froze the leaf tissues in liquid nitrogen, and extracted RNA using a modified CTAB protocol[132]. RNA quality was examined on a 1% agarose gel and RNA concentration was quantified using the Qubit RNA HS assay kit (Invitrogen, USA). Based on the RNA quality and concentration, three samples per time point were picked for sequencing. 2 µg of total RNA was used to construct stranded RNA-seq libraries using the Illumina TruSeq stranded total RNA LT sample prep kit (RS-122-2401 and RS-122-2402). Multiplexed libraries were pooled and sequenced on an Illumina NovaSeq6000 with 2 × 150 bp paired-ends.

**Differential expression analysis**. RNA-seq reads were mapped to the combined nuclear and chloroplast genome using HISAT2[77]. Stringtie[62] was used to assemble transcripts and estimate transcript abundance. A gene count matrix was produced using the included prepDE.py script. We imported gene count data into the DESEQ2 package in R[133] for read normalization using its median of ratios method as well as identification and removal of outlier samples using multidimensional

scaling. A single outlier sample from each of six time points (1, 4, 7, 10, 13, and 19 h) was removed from the final dataset. The resulting dataset was used to analyze temporal gene expression patterns in the R package maSigPro[134]. Using maSigPro, genes with significantly differential expression profiles were identified by computing a regression fit for each gene and filtered based on the associated p-value ($p < 0.001$).

**Identification of CAM-associated genes.** CAM-associated gene identification was accomplished by a combination of functional annotation, homology assessment, and differential expression analysis. Initially, genes previously identified as being involved in the CAM pathway in terrestrial plants were identified using their functional annotations assigned by eggNOG-mapper v2[61] according to sequence similarity. Next, additional putative CAM photosynthesis genes were identified from Orthofinder results if they belonged to any group containing genes identified in the previous step or a group having known CAM-associated genes from *Ananas comosus*[9], *Kalanchoe fedtschenkoi*[10], and/or *Sedum album*[8]. Finally, genes identified in the previous steps were submitted to differential expression analysis to determine whether or not they showed TOD expression. Thus, genes were considered to be "CAM associated" if they exhibited homology to known CAM photosynthetic genes in terrestrial CAM species and cycled in *I. taiwanensis* in a TOD manner.

**HAYSTACK global cycling prediction.** Genes with mean expression across all the time points below 1 TPM were considered "not expressed" and filtered prior to cycling prediction with HAYSTACK (https://gitlab.com/NolanHartwick/super_cycling)[33]. HAYSTACK operates by correlating the observed expression levels of each gene with a variety of user specified models that represent archetypal cycling behavior. We used a model file containing sinusoid, spiking traces, and various rough linear interpolations of sinusoids with periods ranging from 20 to 28 h in one-hour increments and phases ranging from 0 to 23 h in one-hour increments. Genes that correlated with their best fit model at a threshold of $R > 0.8$ were classified as cyclers with phase and period defined by the best fit model. This threshold for calling cycling genes is based on previous validated observations[8,33,34,135]. We also validated this threshold by looking at the cycling of known circadian clock genes (Fig. 5).

**ELEMENT cis-regulatory elements analysis.** Once cycling genes in *I. taiwanensis* were identified, we were able to find putative cis-acting elements associated with TOD expression. Promoters, defined as 500 bp upstream of genes, were extracted for each gene and processed by ELEMENT (https://gitlab.com/salk-tm/snake_pip_element)[33,136,137]. Briefly, ELEMENT generates an exhaustive background model of all 3–7 k-mer using all of the promoters in the genome, and then compares the k-mers (3–7 bp) from the promoters for a specified gene list. Promoters for cycling genes were split according to their TOD expression into "phase" gene lists and k-mers that were overrepresented in any of these 24 promoter sets were identified by ELEMENT. By splitting up cycling genes according to their associated phase, we gained the power to identify k-mers associated with TOD-specific cycling behavior at every hour over the day. Our threshold for identifying a k-mer as being associated with cycling was an FDR < 0.05 in at least one of the comparisons. The significant k-mers were clustered according to sequence similarity (Fig. 6).

**Promoter motif identification.** Core CAM genes with significantly differential diel expression profiles (as identified in maSigPro) including β-CA, PEPC, PEPCK, ME, MDH, and PPDK were selected for motif enrichment analysis. Enriched motifs were identified relative to a background consisting of non-cycling paralogues of photosynthetic genes using the AME utility[138]. Promoters were searched for known motifs from the *Arabidopsis* promoter binding motif database[139] with FIMO[140].

**Reporting summary.** Further information on research design is available in the Nature Research Reporting Summary linked to this article.

## Data availability
A reporting summary for this Article is available as a Supplementary Information file. Additional data supporting the findings of this work are available within the paper and its Supplementary Information. All the raw sequences generated by this study have been deposited in the NCBI Sequence Read Archive under the BioProject PRJNA735564. Genome assembly and annotation are available at https://genomevolution.org/coge/GenomeInfo.pl?gid=61511. Source data are provided with this paper.

## Code availability
Sequence alignments, tree files, and custom scripts can be found at GitHub [https://github.com/dawickell/Isoetes_CAM].

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

## Acknowledgements

We would like to thank Karolina Heyduk, Peter Schafran, and Arthur Zwaenepoel for their advice and support regarding various aspects of this project; Tai-Chung Wu, Wen-Yuan Kao, Ta-Chun Lin, and Chun-Neng Wang for their help on time-course experiments. J.d.V. is supported through funding from the European Research Council (ERC) under the European Union's Horizon 2020 research and innovation programme (grant no. 852725; ERC Starting Grant 'TerreStriAL'), and MAdLand (DFG priority programme 2237; VR132/4-1). A.D.A. and A.D. are grateful for being supported through the International Max Planck Research School (IMPRS) for Genome Science.

## Author contributions

D.W., L.-Y.K., T.P.M. and F.-W.L. coordinated the project. Y.-M.H. provided the plant materials. L.-Y.K. carried out the time-course experiment and nucleic acid extraction. T.P.M. and F.-W.L. sequenced and assembled the genome. H.-P.Y. and F.-W.L. annotated the genome. D.W. assembled the plastome and profiled RNA-editing. D.W. circumscribed gene families and examined genes related to stomata and root development. A.D.A., I.I., A.D., S.d.V. and J.d.V. characterized lignin biosynthesis genes. D.W., Z.L. and M.S.B. carried out WGD analysis. D.W. analyzed expressions of the CAM pathway genes. N.T.H. and T.P.M. carried out HAYSTACK and ELEMENT analyses. D.W., T.P.M. and F.-W.L. synthesized and wrote the manuscript.

## Competing interests

The authors declare no competing interests.
