## [Peer Review File · Nature Communications]

Underwater CAM photosynthesis elucidated by Isoetes genomeReviewers' Comments:

Reviewer #1:

Remarks to the Author:

The authors present an extremely well written manuscript on CAM in Isoetes. This paper is exciting for a number of reasons, but particularly for the CAM aspect. The work done by the authors represents an important missing area in CAM genomics, namely the analysis of a non-angiosperm genome in the context of CAM photosynthesis. I have largely minor or stylistic comments; the science is sound.

L56-61 - even in desert adapted species, the prevailing thought is that CAM is an adaptation to photorespiratory stress that would occur due to low water availability. In other words, photorespiration is at the heart of C4 and CAM, but the ecological conditions that promote that photorespiration vary by lineage and by photosynthetic type (high light vs low water in C4 vs CAM, for example).

L68 - I think "remarkable" is a stretch - see comments further re: L240/Fig. 4.

Fig. 1 - My first reaction was that I really loved this figure, but the more I looked at it, the more I feel like these kinds of figures in general don't convey much to the reader. I appreciate that they're kind of "expected" in genome papers, however. Also, lines are typically used to connect syntenic portions of genomes; do the leaves of the Isoetes picture unintentionally do that? I doubt it. But maybe an inner dark circle that separates the plant image from the chromosome plots would be worthwhile.

Is Fig. 2c referring to the q-values mentioned in the text from WhALE? If so, might make that clearer, simply by putting ("q") on the x axis label of the plot.

L169 - I have wondered about this - are extant Isoetes species quite old, or did they arise more recently?

L240/Fig. 4 - I think the framing of this puts too much emphasis on the convergence of an aspartic acid in Kalanchoe and Phalaenopsis. Essentially, only 2 of the four published CAM genomes show this mutation. Yes, it showed an increase in PPC activity in Yang et al., 2017, but so could other mutations. I think the sentence "This lack of sequence convergence between Isoetes and flowering plants could be the result of their substantial phylogenetic distance..." is a bit of an overstep when two other published CAM genomes (pineapple and sedum) do not have that mutation, either.

P347-364 - Is another potential aspect here that Isoetes reverts to C3 when not submerged? I can't remember if this particular species behaves that way. I could imagine that the environmental sensing TFs could play a role here, too, in unique ways not found in other CAM species (though Sedum is also facultative). It would have been really neat to compare to non-submerged leaves in the RNAseq data to get a sense of CAM "on" switches in this lineage (perhaps that's a later study?).

L 393 - Please provide the herbarium accession number.

L403 - Is there a reason for this to be "in house" - can you provide more details and/or post the script to Github?

L452 - Remove "On the other hand" - a little awkward here, no need for it.

L482 - I again encourage authors to make any "custom scripts" available for others. Even if in a not fully perfect state, they can be super helpful for others as a place to start, rather than everyone re-inventing the wheel.

L486 - Unclear what paranome is?

L586 - How many replicate leaf samples per timepoint (for both RNA and titratable acidity)?

Reviewer #2:

Remarks to the Author:

The authors present a high-quality assembly of the *Isoetes* genome and their comparative genomics analysis provides new insights into the evolution of CAM photosynthesis in aquatic plants, which has been overlooked by the previous CAM genomics studies. The manuscript is well written.

Specific comments:

Page 1 lines 34-35: The author mentioned "the existence of more evolutionary paths to CAM than previously recognized". Please add a diagram in the Discussion section to illustrate the alternative paths of CAM evolution.

Page 3 line 101: Please change "94.5% and 91.0% respectively" to "94.5% and 91.0%, respectively,"

Page 3 lines 99-108: Do the stomata of *I. taiwanensis* open during the night and close during the day, following the same TOD pattern as that in terrestrial CAM plants? Can the authors provide the time-course CO₂ uptake data over a 24-hour period?

Page 3 lines 115-116: In general, the gene density is lower the centromeric regions. Can the authors predicted the centromeric regions and then compare the gene density between the centromeric and non-centromeric regions?

Page 5 line 138: in "single peak at $K_s \sim 1.8$ (Fig. 1a)", do you mean Fig. 2a? Also, I suggest the authors to create a four-fold transversion substitution rate (4d_{tv}) plot for *I. taiwanensis* and other plant species used for comparison. In general, 4d_{tv} is better than K_s for assessing ancient duplications.

Page 6 lines 181-182: The diel expression pattern of PPCK in *Kalanchoe* was not plotted correctly in Fig 3i. Please compare it with Fig 4b in Yang et al 2017 Nat Commun 8, 1899 (<https://doi.org/10.1038/s41467-017-01491-7>). Please double check the TOD plots for other *Kalanchoe* genes.

Page 7, Fig 3: Have you identified the genes for malate import/export? What were the criteria used for identifying the CAM-specific copy among the multiple gene copies in *I. taiwanensis*? Also, please list the gene locus of transcript name for each CAM enzyme in the figure legend, for example, PPCK1 (PHOSPHOENOLPYRUVATE CARBOXYLASE KINASE 1), AT1G08650.1 in *Arabidopsis thaliana*.

Page 7 lines 227-232: I suggest the authors to provide TOD protein abundance data for PEPC, which can be easily obtained through standard proteomics analysis. The TOD protein abundance data may better align with the day/night pattern of CAM physiology than the transcript abundance data.

Page 9, Fig 5: I suggest the authors to add circadian clock genes from CAM species (e.g., *Kalanchoe*, pineapple) for comparison.

Page 13 line 441: Can the author explain why there is need remove repeats with homology to plant proteins?

Reviewer #3:

Remarks to the Author:
see attached

Reviewer #4:

Remarks to the Author:

Wickell and colleagues present a high-quality genome of *Isoetes* and study its primary metabolism and diurnal gene expression.

Overall, we find the study interesting and well-written, but we believe much more information can and should be extracted from the data.

Major points:

1) The authors claim that *I. taiwanensis* recruited a bacterial-type PEPC. This conclusion is based on the high, cyclic expression of the gene. I think this statement is too strong, as this is only based on the TPM values of the two genes, which does not take many factors into account, such as e.g., translation efficiency. As the authors point out, more functional and physiological studies are needed to reinforce this observation. Please tone down this statement, in the manuscript (e.g., '*I. taiwanensis* has recruited bacterial-type PEPC' is too strong) and abstract.

2) The authors could expand and strengthen their analyses by e.g. comparing their data to data from other species in a more quantitative manner. For example, Line 348: 'In sum, TOD-specific enrichment of CREs 348 appears to differ significantly from *Arabidopsis*.'. How many of the relevant elements are found in *I. taiwanensis* and other flowering plants? How similar are the circadian expression patterns to other species (see e.g., citation 36). Which other biological processes show diurnal gene expression?

Minor points:

Line 32: Please rewrite this sentence, as it takes some effort to unpack.

Figure 3. Why is the order of the panels c,d,e,f,a,b,g,h,i?

The author mention that they have estimated heterozygosity, but this is not discussed in the paper.

Response to Reviewers' comments

Reviewer #1:

The authors present an extremely well written manuscript on CAM in Isoetes. This paper is exciting for a number of reasons, but particularly for the CAM aspect. The work done by the authors represents an important missing area in CAM genomics, namely the analysis of a non-angiosperm genome in the context of CAM photosynthesis. I have largely minor or stylistic comments; the science is sound.

L56-61 - even in desert adapted species, the prevailing thought is that CAM is an adaptation to photorespiratory stress that would occur due to low water availability. In other words, photorespiration is at the heart of C4 and CAM, but the ecological conditions that promote that photorespiration vary by lineage and by photosynthetic type (high light vs low water in C4 vs CAM, for example).

Response: This is a very good point. We have revised this part of the paragraph to focus on CO₂ limitation and photorespiration. (page 2, line 57-62)

L68 - I think "remarkable" is a stretch - see comments further re: L240/Fig. 4.

Response: We agree. We changed the sentence to: "Furthermore, a case of amino acid sequence convergence in phosphoenolpyruvate carboxylase (PEPC), which catalyzes the carboxylation of phosphoenolpyruvate (PEP) to yield oxaloacetate (OAA), has also been reported among **some** terrestrial CAM plants¹⁰. (page 2, line 69-72)

Fig. 1 - My first reaction was that I really loved this figure, but the more I looked at it, the more I feel like these kinds of figures in general don't convey much to the reader. I appreciate that they're kind of "expected" in genome papers, however. Also, lines are typically used to connect syntenic portions of genomes; do the leaves of the Isoetes picture unintentionally do that? I doubt it. But maybe an inner dark circle that separates the plant image from the chromosome plots would be worthwhile.

Response: Figure 1 has been modified to make it clearer that those are quills (i.e. *Isoetes* leaves), not syntenic blocks. This figure also illustrates that the distributions of genes and repeats are relatively spread out, unlike most flowering plants.

Is Fig. 2c referring to the q-values mentioned in the text from WhALE? If so, might make that clearer, simply by putting ("q") on the x axis label of the plot.

Response: Thanks for catching this! x-axis of Figure 2c has been altered to include (q).

L169 - I have wondered about this - are extant Isoetes species quite old, or did they arise more recently?

Response: While it is estimated that the diversification of modern *Isoetes* has occurred within the last 100 million years, different studies using different data from different species have arrived at considerably different dates for the crown age of extant *Isoetes* (e.g. Larsen & Rydin 2016 vs. Wood et al. 2020). What we do know, is that plants which unequivocally resemble modern *Isoetes* (i.e. shortened corm with highly restricted apical growth) have existed at least since the Jurassic period (Pigg 2001).

We agree with the reviewer that the description of *Isoetes* being “the oldest extant lineage of vascular plants to exhibit CAM photosynthesis” is not precise. We therefore substantially revised the sentence to read: “The CAM in *Isoetes* is unusual for at least two reasons. First, *Isoetes* diverged from other CAM plants more than 300 million years ago and second, *Isoetes* has an aquatic lifestyle.” (page 6, line 167-168)

Larsén, E. and Rydin, C., 2016. Disentangling the phylogeny of Isoetes (Isoetales), using nuclear and plastid data. *International Journal of Plant Sciences*, 177(2), pp.157-174.

Pigg, K.B., 2001. Isoetalean lycopsid evolution: from the Devonian to the present. *American Fern Journal*, 91(3), pp.99-114.

Wood, D., Besnard, G., Beerling, D.J., Osborne, C.P. and Christin, P.A., 2020. Phylogenomics indicates the “living fossil” *Isoetes* diversified in the Cenozoic. *PloS one*, 15(6), p.e0227525.

L240/Fig. 4 - I think the framing of this puts too much emphasis on the convergence of an aspartic acid in *Kalanchoe* and *Phalaenopsis*. Essentially, only 2 of the four published CAM genomes show this mutation. Yes, it showed an increase in PPC activity in Yang et al., 2017, but so could other mutations. I think the sentence “This lack of sequence convergence between *Isoetes* and flowering plants could be the result of their substantial phylogenetic distance...” is a bit of an overstep when two other published CAM genomes (*pineapple* and *sedum*) do not have that mutation, either.

Response: We believe it is still important to examine the convergent aspartic acid substitution in *Isoetes* PEPC given that it was one of the major findings in Yang et al (2017). We do however agree with the reviewer that such substitution is not ubiquitous among CAM angiosperms and we might have been attacking a strawman. To address this, we have (1) specifically stated that the D substitution is absent in *pineapple* and *Sedum* (page 8, line 242), (2) revised the sentence in question as “This lack of sequence convergence between *Isoetes* and **few** CAM angiosperms could be the result of their substantial phylogenetic distance...” (page 8, line 246-247), and (3) revised the next sentence to read: “Alternatively, it is also likely that the substitution is not as important as previously hypothesized, or relevant only in the context of plant-type PEPC” (page 8, line 248-249).

P347-364 - Is another potential aspect here that *Isoetes* reverts to C3 when not submerged? I can't remember if this particular species behaves that way. I could imagine that the environmental sensing TFs could play a role here, too, in unique ways not found in other CAM species (though *Sedum* is also facultative). It would have been really neat to compare to non-

submerged leaves in the RNAseq data to get a sense of CAM “on” switches in this lineage (perhaps that’s a later study?).

Response: *Isoetes taiwanensis* was reported to have a weaker CAM when emerged (i.e. a lower magnitude of acid fluctuation compared to submerged leaves)(Chang and Yang, 1988). However, we were unable to replicate this; in our experiment, both submerged and emergent leaves have equally strong day/night cycles. That being said, we are planning to conduct a comparative RNA-seq study in a known facultative species of *Isoetes* (*I. engelmannii*; Suissa and Green, 2021) to get at this very question!

Chang, Y.-T. and Yang, K.-J. 1988. Studies of CAM-phenomenon of *Isoetes taiwanensis* Devol. BioFormosa 23: 157-166. pdf link.

Suissa, J.S. and Green, W.A. 2021. CO2 starvation experiments provide support for the carbon-limited hypothesis on the evolution of CAM-like behaviour in *Isoetes*. Annals of Botany 127: 135–141.

L 393 - Please provide the herbarium accession number.

Response: The collection number is Kuo4500, which is now added to this sentence. We are still waiting for TAIIF to officially accession the voucher specimen; the current pandemic has slowed down this process significantly.

L403 - Is there a reason for this to be “in house” - can you provide more details and/or post the script to Github?

Response: Thanks for catching this! There is actually no reason for this being “in house”. We used GenomeScope to estimate the genome size and revised the manuscript accordingly. (page 12, line 402-403)

L452 - Remove “On the other hand” - a little awkward here, no need for it.

Response: Removed.

L482 - I again encourage authors to make any “custom scripts” available for others. Even if in a not fully perfect state, they can be super helpful for others as a place to start, rather than everyone re-inventing the wheel.

Response: RNA summary script has been made available on GitHub: https://github.com/dawickell/Isoetes_CAM. (page 18, line 676-678)

L486 - Unclear what paranome is?

Response: Paranome is a term typically used in reference to WGD analysis referring to all paralogues present in a genome (no single copy genes). We have now defined

“paranome” in the methods as “containing all of the paralogous gene copies in the genome”. (page 5, line 135-136)

L586 - How many replicate leaf samples per timepoint (for both RNA and titratable acidity)?

Response: There were five biological replicates per time point for titratable acidity and three replicates for RNA-seq. We have revised this section to make it clearer how many replicates we have. (page 17, line 600 and line 606-607)

Reviewer #2:

The authors present a high-quality assembly of the *Isoetes* genome and their comparative genomics analysis provides new insights into the evolution of CAM photosynthesis in aquatic plants, which has been overlooked by the previous CAM genomics studies. The manuscript is well written.

Specific comments:

Page 1 lines 34-35: The author mentioned “the existence of more evolutionary paths to CAM than previously recognized”. Please add a diagram in the Discussion section to illustrate the alternative paths of CAM evolution.

Response: We thank the reviewer for this suggestion. We have put a lot of thoughts into formulating this diagram. The difficulty however is that we do not know the chronological order leading to CAM in *Isoetes*, nor in any other CAM plants. For example, did the shift in circadian regulation occur first or did the recruitment of bacterial-type PEPC take place first? In other words, we know that the final “assemblies” are different, but have little clue on the order by which each component got assembled. We are therefore hesitant to put forth a highly speculative figure.

Page 3 line 101: Please change “94.5% and 91.0% respectively” to “94.5% and 91.0%, respectively,”

Response: Revised accordingly.

Page 3 lines 99-108: Do the stomata of *I. taiwanensis* open during the night and close during the day, following the same TOD pattern as that in terrestrial CAM plants? Can the authors provide the time-course CO₂ uptake data over a 24-hour period?

Response:

(1) Stomata. Generally, aquatic species of *Isoetes* either do not produce stomata or have non-functional stomata occluded by wax (Sculthorpe 1967, Keeley and Bowes, 1982). For *I. taiwanensis*, which is amphibious, we have observed that stomata are only produced in emergent leaves and absent in submerged leaves. In other words, it

appears that stomatal control is not directly involved in aquatic CAM. We have elaborated this in the Supplementary Notes.

(2) CO₂ uptake. Measuring CO₂ uptake in an aquatic plant is quite difficult and requires specialized equipment. This was however done by Keeley and Bowes (1982), who demonstrated that carbon uptake was dependent on the concentration of CO₂ in the water and was in fact greater during the day when dissolved CO₂ concentration in the water remained constant. However, CO₂ concentration is highly variable in the field and typically quite low during the day. As a result Keeley concludes that: “Estimating the contribution of light *versus* dark CO₂ uptake to the total carbon gain is complicated by the diurnal flux in CO₂ availability under field conditions.”

Sculthorpe, C.D., 1967. Biology of aquatic vascular plants. E. Arnold, London
Keeley, J.E. and Bowes, G., 1982. Gas exchange characteristics of the submerged aquatic crassulacean acid metabolism plant, *Isoetes howellii*. *Plant physiology*, 70(5), pp.1455-1458.

Page 3 lines 115-116: In general, the gene density is lower the centromeric regions. Can the authors predicted the centromeric regions and then compare the gene density between the centromeric and non-centromeric regions?

Response: We were unable to predict centromeric regions based on gene density alone. While our analysis did find some regions containing long tandem repeats, they did not appear to coincide with a notable decrease in gene density. This could be the result of difficulty in assembling highly repetitive centromeric regions.

Page 5 line 138: in “single peak at Ks ~ 1.8 (Fig. 1a)”, do you mean Fig. 2a? Also, I suggest the authors to create a four-fold transversion substitution rate (4dtv) plot for *I. taiwanensis* and other plant species used for comparison. In general, 4dtv is better than Ks for assessing ancient duplications.

Response: Thanks for catching this! We have fixed the figure reference in the text. Additional 4dtv plots for *I. taiwanensis* and other species are now incorporated in Supplementary Fig. 21. 4dtv plots indeed show a stronger signal for the ISTE β event.

Page 6 lines 181-182: The diel expression pattern of PPCK in *Kalanchoe* was not plotted correctly in Fig 3i. Please compare it with Fig 4b in Yang et al 2017 Nat Commun 8, 1899 (<https://doi.org/10.1038/s41467-017-01491-7>). Please double check the TOD plots for other *Kalanchoe* genes.

Response: We thank the reviewer again for spotting this discrepancy. We have changed the plot in question and double checked all others to ensure that they are accurate.

Page 7, Fig 3: Have you identified the genes for malate import/export? What were the criteria used for identifying the CAM-specific copy among the multiple gene copies in *I. taiwanensis*?

Also, please list the gene locus of transcript name for each CAM enzyme in the figure legend, for example, PPCK1 (PHOSPHOENOLPYRUVATE CARBOXYLASE KINASE 1), AT1G08650.1 in *Arabidopsis thaliana*.

Response:

(1) Malate transport. We were unable to identify a particular transport protein associated with CAM. While ALMTs have been shown to cycle in other CAM plants they do not show strong TOD-specific expression in *Isoetes taiwanensis*. This is now discussed in Supplementary Notes.

(2) CAM gene identification. CAM specific copies of various photosynthetic genes were primarily identified using expression data (i.e. photosynthetic genes that show cycling expression throughout the day were assumed to be associated with CAM). We have added a section to methods detailing how “CAM associated” genes were identified in *Isoetes taiwanensis*. (page 17, line 624-635)

(3) Gene ID. A new supplementary table (Supplementary Table 3) has been added to list all the transcript/locus ID plotted in Figs. 3 and 5, as well as in Supplementary Figs. 26 and 30.

Page 7 lines 227-232: I suggest the authors to provide TOD protein abundance data for PEPC, which can be easily obtained through standard proteomics analysis. The TOD protein abundance data may better align with the day/night pattern of CAM physiology than the transcript abundance data.

Response: While we agree that protein abundance would be preferable (particularly in the case of bacterial-type PEPC), we do not currently have access to the plants used for this study due to the current COVID situation in Taiwan. Unfortunately the co-authors there will not have full access to the laboratory for the foreseeable future making these sorts of tests unfeasible in the near term.

Page 9, Fig 5: I suggest the authors to add circadian clock genes from CAM species (e.g., *Kalanchoe*, pineapple) for comparison

Response: We have added a supplementary figure (Fig. 30) showing the same clock genes we discuss in the body of the paper in several CAM species. There are two reasons we originally chose not to include this figure in the main text or supplement: 1) None of the CAM plants surveyed to date appear to possess unusual expression patterns of highly conserved clock genes such as PRR1 or GI. 2) Several CAM species possess multiple copies of these genes and while they all exhibit similar cycling expression (as you will see in our supplemental figure) it makes for messy comparisons. Due to these factors, we chose to compare *Isoetes* to *Arabidopsis* in Fig. 5 as it provides an easy and effective basis for comparison, while the full comparisons are made available in Supplementary Fig. 30.

Page 13 line 441: Can the author explain why there is need remove repeats with homology to plant proteins?

Response: *Isoetes* genome is rife with genes encoding pentatricopeptide repeat proteins, which sometimes got incorrectly classified as repeats by RepeatModeler. Therefore this step is necessary to “rescue” these genes from the repeat database. We have clarified this in the methods section as: “ The *I. taiwanensis* genome contains a high number of pentatricopeptide repeat genes which are often misclassified as repetitive elements in the genome. Thus, in order to identify and remove repeats with homology to plant proteins...”

Reviewer #3:

Line 27: Perhaps a better way to describe this phenomenon that ties them together is that CAM evolved to provide daytime CO₂ when it is limiting;., either due to stomatal closure in terrestrial arid environments or underwater in oligotrophic aquatic environments.

Response: This is a very good point. We have revised the introduction to focus on CO₂ limitation and photorespiration. (page 2, line 57-62)

Line 59-61: CO₂ availability is the driver in both aquatic and terrestrial CAM plants; in terrestrials it is the result of daytime stomatal closure, in aquatics it is due to low ambient CO₂

Response: We thank the reviewer for pointing this out. We have altered this section to better reflect the fact that CAM is primarily a carbon-concentrating mechanism. The causes of reduced CO₂ availability are merely different. (page 2, line 57-62)

Line 84: it would be good to give authorities for species names

Response: Added name authority (DeVol).

Line 86-89: to what extent have different trajectories been observed within terrestrial CAM plants?

Response: Here we are not so much trying to draw attention to the different trajectories of CAM evolution within terrestrial angiosperms. This has been thoroughly addressed in other papers. We have reworded the sentence to make it clearer/more specific as to the aims of this paper. (page 3, line 89-90)

Line 115-119: this is a very long sentence

Response: We have divided into two separate sentences. (page 3, line 117-121)

Line 120: spore bearing plants or non-seed plants might be a better term

Response: We appreciate this suggestion, but we believe the term “seed-free plants” has been more popularly used in the literature. For example, a recent review on seed-free plant genomics by Szovenyi et al (2021).

Szövényi, P., Gunadi, A. & Li, F.-W. Charting the genomic landscape of seed-free plants. *Nat Plants* 7, 554–565 (2021).

Line 246-248: Lack of convergence in the pathway to CAM is fundamental to the concept of convergence; similar outcomes come about from different pathways

Response: The convergent substitution of PEPC reported by Yang et al (2017) would actually argue that similar outcomes could come from similar pathways. Whether or not this falls under convergent evolution or parallel evolution is perhaps beyond the scope of this paper. Nevertheless, as Reviewer #1 pointed out that the significance of this PEPC substitution is open for interpretation, we have revised this section to reflect the uncertainty. (page 8, line 246-250)

Reviewer #4:

Wickell and colleagues present a high-quality genome of *Isoetes* and study its primary metabolism and diurnal gene expression. Overall, we find the study interesting and well-written, but we believe much more information can and should be extracted from the data.

Major points:

1) The authors claim that *I. taiwanensis* recruited a bacterial-type PEPC. This conclusion is based on the high, cyclic expression of the gene. I think this statement is too strong, as this is only based on the TPM values of the two genes, which does not take many factors into account, such as e.g., translation efficiency. As the authors point out, more functional and physiological studies are needed to reinforce this observation. Please tone down this statement, in the manuscript (e.g., '*I. taiwanensis* has recruited bacterial-type PEPC' is too strong) and abstract.

Response: We have toned down the statements in various places of the manuscript. (line 31, 215, 249, and 379)

2) The authors could expand and strengthen their analyses by e.g. comparing their data to data from other species in a more quantitative manner. For example, Line 348: 'In sum, TOD-specific enrichment of CREs appears to differ significantly from *Arabidopsis*.'. How many of the relevant elements are found in *I. taiwanensis* and other flowering plants? How similar are the circadian expression patterns to other species (see e.g., citation 36). Which other biological processes show diurnal gene expression?

Response: We have now added three new sections to the Supplementary Notes, along with three new Supplementary Tables and Figures, to provide a much more detailed summary of our findings on TOD expression in *I. taiwanensis*. These include GO enrichment analysis as well as comparisons with other land plants (Arabidopsis, Selaginella, and Physcomitrium).

Minor points:

Line 32: Please rewrite this sentence, as it takes some effort to unpack.

Response: Rephrased as: “Notably, *Isoetes* may have recruited the lesser-known “bacterial-type” PEPC, along with the “plant-type” exclusively used in other terrestrial CAM and C4 plants for carboxylation of PEP.” (page 1, line 31-33)

Figure 3. Why is the order of the panels c,d,e,f,a,b,g,h,i?

Response: It is our attempt to reconcile the way the plots are most easily interpreted (i.e. separated according to light and dark reactions) with the order in which they are mentioned in the text. We realize it is a little confusing but it seemed to be the best option instead of rearranging the figure itself or breaking it into multiple figures.

The author mention that they have estimated heterozygosity, but this is not discussed in the paper.

Response: Thanks for pointing this out! Heterozygosity was briefly mentioned in the previous version of the manuscript but was removed because we did not feel the estimate is relevant (nor accurate). We have removed the heterozygosity part from the method.

Reviewers' Comments:

Reviewer #1:

Remarks to the Author:

I thank the authors for their detailed responses to reviewer comments. I have no reservations with the work from a science standpoint, and again I complement the authors on a well-written manuscript. One slight grammatical thing I noticed (below), but otherwise well done.

L89 - change "may have charted their own path" to "may have charted its own path"

I would also request authors make newick versions of their gene trees available on GitHub. Some are there, but I couldn't see the gene trees from the supplemental data on GitHub.

Reviewer #2:

Remarks to the Author:

The authors have addressed my comments well. The manuscript has been significantly improved.

Reviewer #4:

Remarks to the Author:

The authors did a good job revising the manuscript.

REVIEWERS' COMMENTS

Reviewer #1 (Remarks to the Author):

I thank the authors for their detailed responses to reviewer comments. I have no reservations with the work from a science standpoint, and again I complement the authors on a well-written manuscript. One slight grammatical thing I noticed (below), but otherwise well done.

L89 - change "may have charted their own path" to "may have charted its own path"

Response: Revised to read: "may have followed a markedly different path than it has in terrestrial angiosperms."

I would also request authors make newick versions of their gene trees available on GitHub. Some are there, but I couldn't see the gene trees from the supplemental data on GitHub.

Response: Newick files are now available on GitHub.

Reviewer #2 (Remarks to the Author):

The authors have addressed my comments well. The manuscript has been significantly improved.

Response: Thank you!

Reviewer #4 (Remarks to the Author):

The authors did a good job revising the manuscript.

Response: Thank you!